# How to Survive in a Totalitarian Regime: Education of Salesians in Slovakia in the Period of Socialism (1948–1989)

Blanka Kudláčová and Andrej Rajský *

Faculty of Education, Trnava University in Trnava, Priemyselná 4, 918 43 Trnava, Slovakia; blanka.kudlacova@truni.sk
* Correspondence: andrej.rajsky@truni.sk

**Abstract:** After the Communist coup in February 1948, all areas of society in Czechoslovakia were indoctrinated by the ideology of Marxism–Leninism. The Christian Churches became the biggest internal enemies of the state, since, especially in Slovakia, they had a strong tradition and numerous representations. The government tracked representatives and members of the Church and controlled all their activities. The activities of religious orders were prohibited, which de facto abolished them. The only possibility of their preservation was to continue the life, education and work of individual orders in secret. Our goal was to examine how the Salesian order survived in this situation; specifically, we focused on the forms of formation and illegal study of the Salesians during the totalitarian regime. Research in these activities is rather demanding, since they could not be documented for security reasons. So-called memoir literature, the oral history method, and private archival sources were used in the research. Despite the fact that it seems that religious orders would not survive in this historical situation, the opposite was true: in the period of persecution, the orders were not only able to survive, but also raised a new generation that ensured their continuity and the continuation of their activities.

**Keywords:** Slovakia; socialism; Catholic Church; Salesian order; education; persecution

## 1. Introduction—Socio-Political Context and the Relationship between the State and Churches after the Second World War

The political orientation of Czechoslovakia towards the Soviet Union was already formed at the end of the Second World War. In December 1943, an alliance treaty was signed between the Soviet leader Joseph V. Stalin and the Czechoslovak president-in-exile Eduard Beneš, making the Soviet Union Czechoslovakia's closest ally. It should be noted that even though we examine the education of Salesians during the period of socialism in Slovakia, it is impossible to ignore the fact that Slovakia was part of Czechoslovakia at that time.[1] During the Second World War, however, they were two separate states. Adolf Hitler, in the context of his war policy, even before the outbreak of the war, forced the Slovak representation to declare an independent Slovak Republic on 14 March 1939. Two days later, on 16 March 1939, Hitler issued the Decree on the Establishment of the Protectorate of Bohemia and Moravia in the occupied Czech territory at the Prague Castle, without participation of the Czech side. This state of affairs lasted throughout the war, i.e., six years. With the liberation of the Czech territory by allied troops in May 1945 and the defeat of Germany together with its ally, the Slovak Republic, both states disappeared and the Czechoslovak Republic was restored.

After the end of the war, regarding the victorious powers, it was clear that Czechoslovakia would be oriented towards the East. The basic features of the post-war regime were included in the Košice government program (orig. Košický vládny program), which was adopted in April 1945 in Košice. The program explicitly determined the pro-Soviet foreign policy orientation and defined the domestic political regime as people's democratic. This type of regime was built by the Soviet Union in all territories that were under its influence after the war. According to Letz (2021), it was a type of authoritarian left-wing regime,

in which the crucial positions of power were held by the communists, and the adjective "people's" was supposed to express the effort for greater participation of the people in the management of society.

Since the Slovak National Uprising (1944), two parties were prominent on the Slovak political scene: the communist and the democratic one.[2] The particularities and historical traditions of Slovakia were reflected in the results of the elections in 1946, when the democratic party achieved a significant victory in Slovakia (63%), while the communist party won in the Czech Republic (more than 40%). It was probably the last post-war uprising of democratic forces in Slovakia, because both external and internal circumstances set the political direction in favour of the communists long before they seized power on 25 February 1948.[3] After counting the votes for the whole of Czechoslovakia, the Communist Party, with Prime Minister Klement Gottwald, became the winning party. It became the holder of the monopoly of power and established a totalitarian regime, which was gradually implemented in practice. The February coup was only a formal act by which Czechoslovakia became part of the Eastern Bloc controlled by the communists. It represented a completion of a long-term process, controlled mostly from the Soviet Union, which had already begun during the Second World War. The adoption of the Constitution of the Czechoslovak Republic on 9 May 1948 in Prague represented a fundamental step in the formation of the communist regime.

The structure of Slovak society at the turn of the 1940s and 1950s was far from being a suitable environment for applying socialism in practice. On the one hand, a majority of the population identified with the Christian religion[4], and on the other hand, the share of the rural population farming their own land outweighed the share of workers, greatly. Thus, systemic changes in society had to be implemented, inevitably through violence and the persecution of people who did not conform to its ideals (Jašek 2019). Already in October 1945, large enterprises, large estates, banks and insurance companies were nationalized. As of 1 March 1947, more than 3000 enterprises, in which 61% of industry employees worked, were nationalized. In agriculture, there occurred major changes in the area of land ownership during this period, particularly through two land reforms. Self-employed peasants began to be persecuted. The rich peasants who continued in independent farming despite the persistent threat were referred to as kulaks by the regime. An intensive propaganda campaign was directed against them, and if that did not help, mental and physical violence was used. Thousands of residents were sentenced to long-term prison terms, sent to labour camps, and expelled from their hometowns. Other forms of coercion included the dismissal of their children from school, or preventing them from studying at university. These forms of violence were used against all citizens who were "inconvenient" for the regime and did not approve of its political doctrine. The legitimacy of the communist coup and the related purges, carried out by the action committees, was enabled by Act No. 231/1948 on the adjustment of certain conditions for the protection of public interests, dated 21 July 1948. Due to its extreme harshness, it became a symbol of the "illegal acts" of the communist regime (Letz 2021, p. 169). The act was a reaction to mass political emigration from Czechoslovakia after 1948, which was qualified as high treason.[5]

The school system became part of the struggle for political power. Education in Slovakia was already nationalized during the Slovak National Uprising (1944), when private and church schools were abolished. It meant the end of plurality in education. The first socialist school act, No. 95/1948, was passed as early as April 1948. The act launched a rapid process of organizational and ideological restructuring of the system of education, and made it a state monopoly with a pro-Soviet orientation. All schools (except universities, military and theological schools) were definitively nationalized and integrated into a unified school system under the control of the state. The content of education was indoctrinated with the principles of Marxism–Leninism very quickly, with the aim of raising "nationally and politically conscious citizens of a people's democratic state, brave defenders of the homeland and devoted supporters of the working people and socialism."[6]

Christian churches and all their activities had a strong tradition in religious Slovakia. After the communists seized power, churches and their representatives became the biggest internal enemy of the state power, the Catholic Church in particular, since more than 80% of the population identified themselves as Catholics. According to Jakubčin (2012, p. 111), "The Church was the only legally operated institution in the communist Czechoslovakia the worldview orientation of which was not in accordance with the official state ideology of Marxism-Leninism, actually, it even defied it." The representatives of state power paid great attention to the work of churches and wanted to prevent their influence on society. A period of persecution of churches and their members began, which brought the churches under the "guardianship of power" through strong political and repressive interventions (Pešek and Barnovský 1997, p. 9). In regard to the Catholic Church, it was not only about individual interventions, but it was a "systematic persecution campaign" aiming at "submission, weakening and disintegration of the Catholic Church" (Kaplan 1993b, p. 14).

The plan to fight the Church was planned in two stages. "The first stage was supposed to include legal changes that would make the clergy dependent on the state. The second stage was supposed to consist in isolation of the Church hierarchy, which would gradually be replaced by priests devoted to the state." (Grajewski 2002, p. 45). The communists wanted to win believers for an alliance to achieve the construction of socialism, and began to create conditions for the gradual abridgement and cessation of churches and, as a result, religion, too. Already at the turn of 1948 and 1949, the regime mapped the situation in individual churches and their personnel systematically. State security (hereinafter ŠtB) created lists of priests, including records of their activities. Negotiations between the bishops of the Catholic Church and the government were finally interrupted in March 1949, when monitoring devices were discovered at a meeting of the Slovak and Czech bishops in Dolný Smokovec (Jakubčin 2012).

After unsuccessful negotiations with the Catholic Church, but also with other churches, representatives of the state power prepared so-called Church Acts. By Act No. 217/1949, The State Office for Church Affairs (orig. Štátny úrad pre veci cirkevné) was established. Its equivalent in Slovakia was The Slovak Office for Church Affairs (orig. Slovenský úrad pre veci cirkevné) which began its activities in November 1949. Priests thus had a double subordination: to the Church structures and to the office for Church affairs, which caused internal tension among them. In October 1949, another Church Act, No. 218/1949, on the material provision of churches and religious societies was passed. The government presented it as generous compensation for the properties taken from the Church, as it included higher salaries for priests. Pursuant to this law, only priests who had the state's approval and took the oath of loyalty to the state could perform spiritual activities.[7] The act meant absolute subordination of priests to the state power and legalized the possibility of eliminating any inconvenient priest, since state approval could have been withdrawn at any time.

Besides passing the Church Acts, further liquidation actions against the churches followed. In an effort to divide the priests, the state power supported the establishment of the so-called Movements of Patriotic Priests (orig. Hnutia vlasteneckých kňazov), who collaborated with the regime. The Government Decree on Theological Faculties No. 112/1950 abolished all diocesan seminaries, and the study of Roman Catholic theology in Slovakia was concentrated solely at the Faculty of Roman Catholic Theology in Bratislava. At the same time, instructions for the lustration of all teachers and seminarians were issued. The regime tried to liquidate the Greek Catholic Church in Slovakia, which belonged under Rome, by joining it forcibly with the Orthodox Church, officially requesting its jurisdiction at the Orthodox Patriarch of Moscow (Action P). In eastern Slovakia, where these churches are largely represented, it caused a great deal of animosity among the believers. In 1950, two interventions against male orders were carried out (Action K1 in April[8] and K2 in May).[9] Security forces occupied all monasteries, and the religious were concentrated in centralized internment monasteries. A similar intervention against female orders was carried out in August 1950 (Action R). Religious sisters and men were either imprisoned

or put to production as manual workers, or they could become civilians. Young religious who had not completed military service yet were taken to technical auxiliary battalions (orig. PTP)[10] and the eldest religious were concentrated in charity houses. Thus, the female and male orders were de facto abolished in Slovakia. In 1950, the regime isolated and interned all the bishops, many of whom were imprisoned for several years; some were sentenced to life imprisonment. In the course of the same year, state guards were placed in episcopal chairs, and the activity of the remaining bishops was eliminated. In the 1950s, several political monster trials[11] took place, the victims of which were forced to confess to fabricated charges, often through psychological and physical torture. A particularly negative role was played by the ŠtB in this period, which became the regime's most feared tool in persecuting citizens and preparing and implementing political processes.[12] All Church associations and magazines were also abolished. According to Mikloško (1991), the communist regime in Czechoslovakia tried to create a so-called national Church, seceded from the Vatican. The state supported only pro-regime priests and their activities, which were constituted by platforms created by the state, e.g., Catholic Action (orig. Katolícka akcia), Peace Movement of Catholic Priests (orig. Mierové hnutie katolíckych kňazov) as well as the weekly Katolícke noviny (tr. The Catholic Papers).[13] Activities outside of the state-controlled movements were considered anti-state activity and they were severely punished. The churches in Slovakia were disrupted and incapable of proper functioning.[14] Grajewski describes the situation as follows: "Evaluation of the Stalinist epoch was tragic for the Church both in the Czech lands and Slovakia. It was a Church without bishops and priests, strictly controlled by the state administration and the clergy who submitted to it." (Grajewski 2002, p. 49). The Catholic Church was pre-emptively preparing itself for the upcoming difficult period, as it could not make decisions on many serious matters and the contact with the Vatican was interrupted. This is also evidenced by the so-called secret faculties, on the basis of which the Church had the right to decide on matters that were otherwise exclusively subject to the Vatican. Their goal was to ensure the activity of the Catholic Church at the time when bishops and Church dignitaries were imprisoned and isolated. The illegal network, which was gradually being formed, ensured rapid dissemination of information and it assisted in the secret ordination of priests later on (Kaplan 1993a, pp. 136–37). The first secret bishop in Slovakia was the Jesuit Pavol Hnilica, who was consecrated by the official bishop Róbert Pobožný in January 1951. In the same year, Hnilica consecrated the Jesuit Ján Chrysostom Korec, who was the greatest moral authority in the secret church, as bishop. Most of the secret bishops were Jesuits, and as secret priests they were basically ordained religious. Hnilica emigrated from Slovakia in 1952 and in 1954 presented to Pope Pius XII and to the Superior General of the Jesuits in Rome a report on the persecution of the Church in Czechoslovakia. The Pope responded to this news with a decree in July 1954, in which he established the exact conditions for the secret consecration of bishops and thus ensured continuous potestas ordinis, i.e., the state when at least two bishops were to be secretly consecrated in Slovakia at the same time. In general, the decree represented the basis for the establishment of secret Church structures in Slovakia, and de facto remained in force until the fall of the communist regime.

## 2. Goal of the Study, Research Methods and Current State of Research

The goal of the study is to examine how the Salesian order survived in the period of socialism under the conditions of persecution of the churches in Slovakia. Specifically, the focus is on the formation and forms of illegal education of new Salesians during the totalitarian regime. It represents a specific phenomenon of religious education in the period of unfreedom that has been examined only partially up to now. The Salesian order was chosen as a representative of orders that operated in Slovakia continuously in the interwar period, during the Second World War and illegally during the socialist totalitarian regime. The Salesians were numerically the largest male religious order in Slovakia in the period under review, and, at the same time, they belonged to the most active church communities that operated during the communist persecution.

Since the formation and education of Salesians could only take place in secret, information on it could not be recorded and documented for security reasons. Their actors were threatened with interrogation, various kinds of persecution, or prison. These activities took place only on the basis of personal contacts and trust (Kudláčová 2020). For this reason, the activities cannot be examined only on the basis of direct archival sources and period documents, but they are examined also indirectly, through memoir literature, samizdat literature, state security records, or newer historical methods, e.g., oral history and narrative biography. In the presented research, memoir literature, the oral history method and archival sources were used.

Considering memoir literature, the authors are the actors, the Salesians themselves, who experienced the period of socialism and published their personal notes or processed information and knowledge they remembered after the fall of the regime. An important source for the research is represented by publications by Ernest Macák (1996, 2000, 2004, 2006), where he processed his notes taken during the years of persecution. Memoir literature, generally depicting persecution of churches and orders in the period of socialism, represents a great source, too, e.g., Ján Chryzostom Korec (1989), František Mikloško (1991), František Vnuk (1995), Ján Milan Dubovský (1998), Ján Šimulčík (2017), Lucia Bačíková (2005), Marián Gavenda (2016). Research conducted by Annamária Adamčíková is another valuable source of information. She conducted interviews with 22 living Salesians who were active in the period of socialism (it concerns mainly the second generation that was active in the 1970s and 1980s; the first generation passed away). The interviews took place between February 2019 and February 2020. Regarding interviews, the authors of the presented study conducted three interviews (Baňovič 2020; Baranyai 2020; Urbančok 2020). The final source of information in the research is represented by the Archive of the Slovak Province of Salesians in Bratislava (orig. Archív slovenskej provincie saleziánov v Bratislave) (raw fund of Jozef Izakovič and the fund of Andrej Dermek), the private archives of Peter Timko SDB and the report on the visitation of Salesians in Czechoslovakia in 1968 (Ter Schure 1969, Archivio Salesiano Centrale Casa Generalizia Salesiana, Rome).

Considering secondary literary sources dealing with the persecution and reconstruction of the life of the Salesian order under socialism in Slovakia, the following authors need to be noted in particular: Peter Timko SDB, Juraj Kováč SDB, Milan Urbančok SDB, Vladimír Fekete SDB, Imrich Tóth SDB, Jozef Bližinský SDB, Peter Lukáčik SDB, Kamila Novosedlíková FMA, Andrej Rajský, Adriana Sarközyová, Annamária Adamčíková. Results of their research that has been rather partial up to now have been published in studies or qualification theses (Lukáčik, Bližinský, Timko, Sarközyová, Adamčíková).

Based on available memoir and secondary literature, the following questions have been posed:

(1) When did the Salesian order start working underground and what were the crucial factors for starting this activity? (2) Who decided, and in what way, on the formation and study of Salesians in the underground? (3) What were the possibilities of the secret study; how was it organized? (4) What was its content; who provided individual subjects? (5) What were the particularities of the illegal education?

## 3. Illegal Study and Formation of New Salesians in Slovakia in the Period of Socialism

The Salesian society[15] in Slovakia has belonged to large religious communities since its inception. The subject of its activity is primarily education of the youth, pastoral care of vocations, pastoral care of the people and missionary work. Currently, Salesians are the largest male religious order in Slovakia; their number is around 200. Since the activities of Salesians were focused mainly on the education of children and youth, the communists considered it one of the most dangerous religious orders. Together with the Jesuits, they belonged to a sample in which the communists observed a reaction in believers when some of their community houses were abolished after 1948 (Gavenda 2016, p. 44).

### 3.1. Salesians in Slovakia—A Short Historical Excursion

The first Slovaks interested in joining the Salesian order left for Italy in 1905 to study. Viliam Vagač, the founder of the Salesian work in Slovakia, was among them. The first Slovak Salesians founded a Slovak foreign grammar school in Italy (in the village of Perosa Argentina) in 1921. Until 1924, approximately 100 Slovak and Czech students studied there (Kubanovič 2019). In 1924, the Salesians in Slovakia opened the first boarding school for religious youth (in Šaštín, later a grammar school) and in 1929, they established a higher school for clerics (in St. Beňadik, novitiate and pedagogical–philosophical studies). In 1934, there were already 220 Salesians in Slovakia, including novices and clerics (Červeň 1998). More than half of them studied in Italy. Foreign study by Salesians in the interwar period was a very good experience and intellectual preparation for qualified work in the developing Slovak society in the interwar period, as well as during the war and in the post-war period. According to Rajský (2020, p. 93), language skills, high-quality theological and philosophical education of the majority of Slovak Salesians, and personal contacts "made it possible to build and maintain parallel structures of formation of adolescents during the socialist period, including later illegal theological studies, which were provided also by professors from own ranks."

In 1935, an independent Czechoslovak province of Salesians was established (after the separation from the Yugoslav province) and the first Czech Salesian, Ignác Stuchlý, became the first provincial. Four years later (1939), the Czechoslovak province was divided and an independent Slovak province was created. Jozef Bokor became its first provincial (then inspector). Bokor worked in an extremely difficult time, both during the Second World War and during the harshest persecution of religious orders in the 1950s, until 1969, when he was succeeded by Andrej Dermek. Dermek held this position until 1981, when he was succeeded by Jozef Kaiser.

### 3.2. The Salesian Society at the Beginning of Socialism

At the beginning of socialism, Salesians worked in several places in Slovakia; in 1948 there were 13 places (Dubovský 1998, p. 14). Regarding the number of Salesians before the dissolution of religious orders in 1950, Paštéka (2000) state that there were 288 of them. The same figure is also declared by Ernest Macák in his Zápisky spoza mreží[16] (tr. Notes from Behind the Bars) (Macák 1996, p. 37). Another figure can be found in Dubovský (1998), who talks about the concentration of 256 Salesians during Action K1 (April 1950), but, apparently, Salesians who were sent by the province to diocesan parishes were not included.

Following the political coup in 1948, the Slovak Salesians were reflecting on their situation and perspective, as Church schools and boarding schools were being gradually nationalized already as of 1945. In 1949, the first religious houses focused on pastoral care for youth were also abolished. In total, around 40 such houses were closed; in the case of Salesians, four of them: in Komárno, Michalovce, Žilina and Trnava (Dubovský 1998, pp. 9–10). For this reason, the provincial Bokor released about twenty Salesians for service in the dioceses[17], which meant that during the Barbarian Night some Salesians were not interned in Šaštín, or later in Podolínec. Due to the worsening situation of the churches, Bokor appointed František Valábek as his deputy, who was supposed to substitute for him in the event of his persecution or imprisonment. He also urged all members of the order to observe the religious Statutes as much as possible and clerics to continue their studies (Zanet 2017, p. 82). In March 1950, the religious houses of Jesuits and Salesians, in which their novices and prospective priests were formed, were nationalized. Kováč states (Kováč 2012) that in the case of Salesians, it concerned the following houses: Galanta-Hody (philosophical-pedagogical training of students), Hronský Sv. Beňadik (novitiate), Šaštín (grammar school) and Sv. Kríž nad Hronom (philosophical and theological training— Higher School of Theology (orig. Vysoká bohoslovecká škola]).

During Action K1 (at the turn of 13 and 14 April 1950), the Salesians were forcibly concentrated at first in Šaštín and, ten days later, they were moved to a centralization

monastery in Podolínec, where friars from various male orders from across Slovakia were concentrated. Among them, there were about 150 young Salesians in formation, 16 young priests of up to 35 years of age, 26 priests between 35 and 55 years of age and 40 fraternity brothers[18] (Macák 1996). The imprisoned Salesians received an offer in Šaštín that first-year novices could go home to their parents, clerics and fraternity brothers could go to civilian life, and priests could leave the order and enter diocesan parishes. However, none of the imprisoned signed this statement. "There were almost three hundred of us in the camp, but no one took a pen, no one rushed to sign it." (Macák 1996, p. 49). The provincial Bokor was transferred to the internment monastery in Želiva in Bohemia, as he was labelled reactionary and uneducable. The internment monastery had a stricter regime than the centralization monastery[19]. After six years of permanent interrogations, the ŠtB arrested him and he was sentenced to four years in prison. In 1960, he was released with poor health and never received state approval for public ministry again.

At the beginning of the persecution, Salesians hoped that it was only a temporary situation and that they would be able to continue their activities in a short time. They were concentrated in Podolínec from April to August 1950. The centralization monastery was abolished in December 1951. On 22 May 1950, approximately 90 of the youngest Salesians, originally grammar school students, were sent from Podolínec for "re-education" in Marxism–Leninism to Kostolná near Trenčín. From there, they were transferred to work on the "Youth Building"[20] (The Youth Dam [orig. Priehrada mládeže] near Púchov) during the summer holidays.

The Salesians created the first system of illegal education already in Podolínec. In his notes, Macák states that after only ten days they started organizing studies, and devoted two to three hours a day to studying. "We take out our books and secretly carry them to work in case there is some extra time left there. We read secretly and also in public . . . We study privately, but the clerics of the pedagogical seminar also organize it. They formed 7-member circles of German, Italian, and Latin. Each circle has its own instructor who is a priest or an older cleric. Each circle meets for an hour in the morning and for an hour in the afternoon" (Macák 1996, pp. 98–99). This is also confirmed by Ján Brichta's statements from 2010[21]. Individual orders organized their studies separately; they did not mix up with each other. The Salesians in Podolínec even created a secret committee of their professors from the abolished theological college in Sv. Kríž nad Hronom[22]. It was led by Andrej Dermek, and its goal was that the Salesians could continue studying theology even under these conditions (Kováč 2012, p. 51). At the beginning of August, the police confiscated all the books they found from the religious. This made studying almost impossible: "They stole armfuls of books from us, they want to terrorize us and starve us mentally. All of us in the camp have a very intimate relationship to books . . . their loss is one of the hardest strokes we have experienced here" (Macák 1996, p. 185). On 4 and 6 September 1950, approximately 120 theologians were transferred from Podolínec for military service in Bohemia, and at the end of September, approximately 40 older theologians were included in the technical auxiliary battalions (hereinafter PTP) (Rajský 2020).

Ernest Macák understood that young Salesians in youth camps and in military service needed support and encouragement so that they would not become discouraged. He escaped from the Podolínec monastery twice in order to secretly establish contacts with young brothers placed in labour camps and help them remain in the Salesian vocation. The first escape was only "explorative" and he returned back to the camp. However, the second escape (10 October 1950) was definitive, with the permission of the deputy provincial Valábek. Valábek entrusted Macák with the care of young Salesians and also with organizing their escapes abroad.[23] "In these extraordinary times, Father Valábek had the faculties of our superiors and I think also of the Holy See. Before I escaped from Podolínec, I met him several times after dark in the attic of the camp... He agreed, gave me his advice, and I escaped from the camp with his blessing" (Macák 2004, p. 12). Regarding the organization of escapes, Macák teamed up with Titus Zeman, who was at the parish in Šenkvice near Trnava at the time. Zeman organized three expeditions, two successful;

the third one was unsuccessful.[24] After the third, Zeman and 19 other people involved in the escape were sentenced to 290 years in prison, Zeman to 25 years. Macák continued to organize escapes and teamed up with František Reves. Together they made six expeditions between October 1951 and July 1952 (Adamčíková 2020, pp. 48–49). Reves had to hide, and in the spring of 1952, he also fled[25]; Macák was arrested in 1952. Thus, escapes over the Morava River were stopped. Later, there were only individual emigrations.

Young Salesians spent the years between 1950 and 1953/54 in youth building sites and PTP camps. The goal of the regime was to isolate older friars (often in prisons) and to force or allure the younger ones to leave the order. Staying at youth building sites in mixed collectives with girls and women and later in difficult working conditions of the PTP was intended to discourage young Salesians from religious life or put them in compromising situations. In a PTP, the situation with study also changed; it was no longer possible to follow the direct guidance of superiors as it was in Podolínec, and professors were not available. Their study turned into self-study and personal consultations with authorized teachers. Jozef Izakovič recalls his studies in the PTP as follows. "When we took the steel forgings out of the acid and loaded them into large cleaners, in which, making use of steel sand, they were meant to get rid of forge scale dissolved by the acid, which always took a long time, we learned the texts of the Holy Mass by heart and rehearsed its rites. We took theology exams during occasional visits of Don Ernest Macák, who escaped from Podolínec and devoted all the moments of his still uncertain freedom to visiting and encouraging the brothers and helping them in every possible way" (testimony of Izakovič, in Bačíková 2005, pp. 69–70).

Considering the conditions of the 1950s, as described above, Salesians had to decide how and to what extent they would participate in the continuation of the activity of the order and in what way the formation and education of Salesians would take place. Salesian candidates had three options for pursuing philosophical and theological studies: (1) to study abroad (where they had to flee first), (2) to study at the Roman Catholic Faculty of Theology in Bratislava, where it was difficult to enrol due to the so-called numerus clausus[26], and the faculty was influenced and controlled by the regime, (3) illegal studies (in parallel with studies at another university or employment). Considering the goals of the presented research, illegal studies in Slovakia represent the major interest. The results of the findings were arranged chronologically, according to the terms of office of individual Salesian provincials.

### 3.3. Illegal Study of Theology in the Period of the Provincial Jozef Bokor (1950–1968)

As a result of the abolition of the Salesian Theological College in Sv. Kríž nad Hronom and the abolition of religious orders, Salesians could not complete their studies properly, and they were waiting for the next opportunity to receive the sacrament of priesthood. These were deacons working in PTPs. They could receive the so-called Christmas holiday for good results at work, which they used for secret priestly ordinations. This opportunity was used by the deacons of the fourth year: Pavol Čerňanský, Rudolf Granec, Stanislav Sloboda, Ján Szabadoš (Adamčíková 2020, p. 18). Consent was granted by the deputy provincial Valábek. Ján Beňo and Miroslav Kysela, who could not go on Christmas holiday, received the sacrament of priesthood from Bishop Róbert Pobožný in Rožňava. Despite not completing their theological studies, these priests were granted an exemption to be ordained and continued their studies secretly after priestly ordination. In the case of younger students of theology, their education was changed to individual education provided by older priests, who were entrusted with this by Valábek. Between 1950 and 1952, Macák coordinated their studies. After his second escape from Podolínec, he started visiting them directly at the Youth Dam and kept in contact with them at least once every four months. He encouraged them to persevere in their profession and try to study theology in their free time (Macák 2000, pp. 180–94).

Macák prepared the first concept of complete secret philosophical and theological studies in the underground. "In the school year 1951/52, together with several theologians

we tried to organize a course in philosophy, which preceded the study of theology. For the school year 1952/1953, I prepared philosophy and theology courses for other brothers. In the autumn of 1952, some theologians were to return from forced military service and there were several very gifted brothers among them. We wanted to start the novitiate as well. The foundations of religious life under the new conditions were thus outlined. We were collecting our first valuable experiences ploddingly. It appeared that it would be possible. It seemed realistic that religious life would be possible even without religious houses, that it would be possible to establish religious houses on a territorial basis, i.e., religious brothers living alone will have their superior in the district or in the city. I have already had an approval for the novitiate and studies from Father František Valábek[27], whom I met at the beginning of spring 1952, when he was in hospital in Semily in Bohemia" (Macák 2004, p. 13; 2006, p. 295). The expected length of study was six years; two years studying Christian philosophy, and then theology, one subject at a time. However, given the challenging conditions, limited meetings with teachers, lack of professional literature[28] and ability to perform other duties of theology students, the speed and the length of their studies differed. Dermek recalls, "It was so miserable, it was at stations, in parks, at cemeteries, where we used to meet to lecture, examine, explain" (Dermek 1999, p. 30). Beňo called this form of study "guerrilla", far from the classical institutional education typical of older brothers (Beňo 1994).

Macák was very busy with organizing, yet he gave an introductory lecture on Christian philosophy and its meaning. Subsequently, the theologian Jozef Kubička, who was the only one to graduate in philosophy, continued (Macák 2000, pp. 197–202). Several Salesians who started secret studies were imprisoned after Titus Zeman's third unsuccessful expedition. This prolonged their studies because the conditions in the prison were not suitable and it was not possible to take exams. For example, Alojz Masný studied for 15 years (Masný 2014). In September 1952, Macák was arrested. After imprisonment and torture, in December 1952 he began to pretend to be insane.[29] In April 1955, he was released from compulsory treatment to his parents' home, where he lived for the next 13 years pretending to be mentally disturbed and secretly wrote. In 1968, he was able to go into exile in Rome with his brother Ľudovít. From 1976 to 1985 he was the director of the Slovak grammar school in Rome, after the fall of the regime he returned to Slovakia and in 1990 he restored the Salesian grammar school in Šaštín.

In the 1950s, the formation and education of young Salesians was provided mainly by: Ján Beňo (organized the secret activity in Zvolen), Viliam Vagač (organized the secret activity in Bratislava and mediated secret priestly ordinations of Salesians with the secret bishops Hnilica and Korec), Jozef Kubička (took over the secret activity in Bratislava after the arrest of Vagač), Miroslav Kysela (organized the secret activity in Žilina), Ján Mikes (organized the secret activity in Bratislava), Jozef Izakovič (took over the secret activity in Bratislava after Kubička). However, several of them were sentenced: in 1952, Vagač to eighteen years in prison; in 1956, Jozef Bokor, František Valábek, Ján Mikes, Ján Beňo and Alojz Masný to three years in prison (cf. Rajský 2020; Adamčíková 2020).

The admission of the first Salesian to the secret novitiate represented a breakthrough event. It was Ivan Gróf, who completed the first months of the novitiate under the novice master Jozef Kubička, who was later succeeded by Jozef Izakovič (he held the position between 1954 and 1978). On 12 December 1954, in his apartment in Trnávka, Ivan Gróf took his first religious vows for three years administered by Don Ján Beňo.[30] Subsequently, Gróf completed philosophical studies, lasting approximately one year, and continued with exams in dogmatic and moral theology. This was an innovation, since the study of theology took place after taking lifetime vows. This model was later approved by the competent Salesians in Turin and it helped to maintain the Salesian society. In 1959, Gróf received the sacrament of priesthood together with Jozef Kosmál. Other members were gradually being admitted to the novitiate. Izakovič called this method of novitiate "individual and 'peripatetic'—marching. For security reasons, I always met individual novices at different places and the novices even did not know each other..."[31] In his notes, he states, "Until 1968,

the entire formation community consisted mostly of only two people: the formed and the forming. We lived in such conditions that every extra eye and ear were dangerous, because they were connected to the mouth, and the mouth often due to carelessness, imprudence, fear and weakness did not say only those things it did not know" (statement of Izakovič, in Bačíková 2005, p. 76).

After the difficult 1950s, marked by the influence of Stalinism, political relaxation began in Czechoslovakia in the 1960s. One of the reasons was the appointment of Alexander Dubček as the first secretary of the Central Committee of the Communist Party of Czechoslovakia (orig. Komunistická strana Československa) (hereinafter KSČ) and his "socialism with a human face". The Czechoslovak bishops demanded autonomy in the leadership of the Church from the government, and additionally, restoration of religious life. In 1960, on the anniversary of the liberation of Czechoslovakia, President Antonín Novotný announced an amnesty that applied exclusively to political prisoners. Many prisoners were released[32], including several Salesians. This was followed by amnesties in 1962, 1965 and 1968, when the last political prisoners were released. After being released from prison, the Salesians tried to orient themselves in the new situation. Dermek writes, "I was 46 years old at the time, I was conditionally released from prison, I had one set of clothes and six hundred crowns in my pocket as a balance from a four-year account, for four years of work. And a big question mark in front of me: And what now?" (Dermek 1999, p. 16). Former political prisoners could only work as labourers, e.g., Dermek worked as a storekeeper and supplier, Beňo as a lumberman.

### 3.4. Illegal Study of Theology in the Period of the Provincial Andrej Dermek (1968–1981)

In the spring of 1968, superiors of all religious orders in Czechoslovakia created the Secretariat of Religious Societies and elected Andrej Dermek as the representative for Slovakia. Provincial Jozef Bokor died on 1 April 1968. It was necessary to appoint a new provincial and to reorganize the scattered and rather horizontally networked province. Thanks to the political release, Don J. G. Ter Schure, the superior of the Salesians in Rome for Central Europe, was able to visit Slovakia in May 1968. The visit was arranged by Dermek. During the visit, Ter Schure could observe the state of Salesians in Slovakia, contained in a detailed report available in the central archive in Rome (Ter Schure 1969, ASC F969). Andrej Dermek was appointed the new provincial of Salesians in Slovakia, František Valábek the vicar and Ján Vizváry the economist of the province. These three, together with Beňo, Kysel and Izakovič, formed the Provincial Council that was supposed to "admit candidates to novitiate and allow for vows as well as ordination, which has been carried out without it due to absence of the Council" (testimony of Izakovič, in Bačíková 2005, p. 78). Dermek contacted all his fellow brothers, confirmed the so-called territorial communities, which were thereby legalized, and determined the processes of management and communication with superiors. Some Salesians expected to return to standard religious life. However, the change in the political situation after the invasion of the Warsaw Pact troops in August 1968 made it necessary to retreat into the previous position.

In April 1969, Dubček was removed from his position, and in May of the same year, the Communist Party of Czechoslovakia approved legislation that marked the start of the process of the so-called normalization.[33] It included new anti-Church measures. Dermek, as the new provincial, was an employee of the Salesian Society; he had to leave for civilian employment again in 1970. The State Security dissolved some Salesian communities. This external pressure was multiplied by internal arguments in the Provincial Council, which became divided. On the one hand, there were more moderate views represented by elder Salesians (Dermek, Valábek, Vizváry, Kysel); on the other hand, there was a more radical wing, represented by Beňo and Izakovič. Differences in opinion were related to the concept of religious life under the changed normalization conditions and caused constant tensions (Lukáčik 2005, p. 122).

If the formation and method of study until 1968 can be called peripatetic or individual, and it lacked the community dimension typical of Salesians, from the 1970s on, Salesians

began to form small, so-called ambulatory communities. These were mostly two to four Salesians living together in one apartment. After completing the novitiate, they could continue their philosophical and theological studies, mostly in parallel with their studies at another university or with employment. The quality of the studies did not reach the level of regular studies and was largely dependent on the responsibility and interest of the candidate. "Individuals who cared about the depth of scientific and spiritual knowledge had the opportunity to study individually in depth—it depended on personal motivation".[34] Since individual courses were not studied in parallel, but one after another, the length of study varied, ranging from 6 to 9 years. The courses were taught by several lecturers, who were also responsible for providing the students with study literature.[35] Table 1 presents courses and lecturers, as well as a list of students who completed their studies and graduated during the period, in the provincial Dermek.

**Table 1.** List of courses, lecturers and students in the 1970s and early 1980s.

| Course | Lecturers between 1965 and 1983 | Students and Years of Study |
|---|---|---|
| Philosophy | Izakovič, Študent, Sloboda | |
| Old Testament | Dermek, Beňo | |
| New Testament | Dermek, Beňo, Obonya | |
| Fundamental Theology | Pobiecky, Vašek, Šebo | |
| Dogmatics | Valábek, Masný, Pobiecky, Gróf, Šebo | Jozef Gánovský, 1965–1972 Štefan Urban, 1970–? Jozef Daniel Pravda, 1970–1976 Ján Zauška, 1972–1980 Milan Ficík, 1976–1984 Jozef Dömeny, 1974–1981 Jozef Gibala, 1976–1982 Jozef Lančarič, 1972–1980 Vladimír Fekete, 1976–1983 |
| Pastoral Theology | Beňo, Študent (connected with formation) | |
| Homiletics | Beňo (connected with formation) | |
| Church History | Dömeny, Študent, Dermek | |
| Moral Theology (general) | Valábek, Butaš, Šebo, Masný | |
| Moral Theology (special) | Valábek, Butaš, Masný Bazala, Butaš | |
| Canon Law | ? | |
| Liturgics | Brunclík, Chudý, Izakovič | |
| Asceticism, Mysticism | Masný, Beňo[36] | |

Despite the difficult conditions, Salesians tried to capture the development of post-conciliar theological thinking, and regular studies were amended with occasional study weekends or weekly study stays during this period. They took place in Slovakia, Poland or Hungary. Their organization was demanding; the problem was mainly securing days off for those Salesians who worked. In Hungary, these stays took place in Esztergom, where Slovak professors from Rome (e.g., Slovak biblical scholar Jozef Heriban) delivered lectures. Initially, study stays were also held in Poland; later, due to simpler logistics, Polish priests came to lecture in Slovakia (Fekete's testimony, in Bačíková 2005, p. 93). Adamčíková (2020) states that one of the initiators of these stays was Július Obonya.

*3.5. Illegal Study of Theology during the Years of the Provincial Jozef Kaiser (1981–1993)*

In 1980, Jozef Kaiser was appointed the provincial, while Dermek was asked to remain in office for one more year. During this year, Kaiser acquired much necessary knowledge (Lukáčik 2005, p. 43). In 1982, formation was divided into two formation communities:

the East and the West. This reflected the increase in the number of applicants interested in joining the Salesian Society. Ivan Gróf was responsible for the western part of Slovakia (from Bratislava to Banská Bystrica) (until 1987), Jozef Šebo for the eastern part (Adamčíková 2020). This practical measure made commuting easier for both the lecturers and students. Individual lecturers were determined by the provincial Kaiser, probably after agreement with the Provincial Council (see Table 2). The lecturers continued their individual approach with the students, because they still did not have the same possibilities for studying, especially regarding time. Those who studied at a parallel university had more time; those who worked had less time.

**Table 2.** List of courses, lecturers and students in the 1980s and early 1990s.

| Course | Lecturers between 1981 and 1992 Western Slovakia/Eastern Slovakia | Students and Years of Study |
|---|---|---|
| Philosophy | Gróf, Študent/Lančarič, Sloboda | |
| Old Testament | Dermek, Dömeny/Marušiak, Bližinský | |
| New Testament | Dömeny, Obonya/Fekete | |
| Fundamental Theology | Gróf, Sloboda/Šebo | *Western Slovakia:* Vincent Feledík (1981–1988) Jozef Bago (1985–1992) Cyril Gajdoš (1985–1992) Milan Urbančok (1986–1992) |
| Dogmatics | Bazala, Gróf, Sloboda/Šebo | |
| Pastoral Theology | Sloboda, Brunclík /Šaštín, Kutarňa, Šebo, Sloboda | |
| Church History | Študent/Študent, Šaštín, Pravda | |
| Moral Theology | Brunclík/Šebo, Butaš, Bazala, Brunclík | *Eastern Slovakia:* Anton Červeň (1981–1988) Vincent Macejko (1986–1992) Pavol Seman (1986–1992) Peter Bešenyei (1987–1992) Jozef Luscoň (1988–1991) |
| Liturgics | Pravda/Pravda, Šaštín, Valábek | |
| Pedagogy | Kutarňa/Kutarňa, Šaštín | |
| Catechetics | Kutarňa, Šaštín/Kutarňa, Šaštín, Šebo, Sloboda | |
| Canon Law | Ganovský/? | |
| Asceticism, Mysticism | Sloboda/Sloboda, Šaštín, Šebo[37] | |

In the 1980s, changes in political development occurred in the countries of Eastern Europe. The key change was the rise of Mikhail Gorbachev, who began to implement extensive economic reforms and the democratization of society, known as perestroika and glasnost. In Czechoslovakia, the reaction to these changes was slow. However, the poor economic situation, technological lagging behind the West, poor environmental conditions and non-compliance with human rights and religious freedoms caused increasing dissatisfaction among the population. In the second half of the 1980s, all anti-regime groups, whose activities began to intertwine, were activated. The most numerous part of the dissent was the Christian dissent, which began to emerge more and more frequently in public in the 1980s. The first mass public demonstration was a pilgrimage in the Moravian town of Velehrad in 1985, which was attended by around 200,000 young people. In the 1980s, the phenomenon of pilgrimages played a very important role in open expressions of faith and protests against the violation of religious freedoms in Slovakia. The number of pilgrims gradually increased; in 1988, 250,000 believers, mostly young people, took part in the pilgrimage to Levoča (Letz 2021). The petition of the Moravian Catholics, Podnety

katolíkov k riešeniu situácie veriacich občanov v ČSSR (tr. Suggestions of Catholics for the Solution of the Situation of Religious Citizens in the Czechoslovak Socialist Republic) represented an important strengthening of the self-confidence of the believers, too. It was signed by over half a million people, including almost 300,000 Slovaks (Šimulčík 2017). The candlelight demonstration in Bratislava of 25 March 1988 (the so-called Bratislava Good Friday) is considered the most significant upheaval. The peaceful demonstration of citizens for religious and civil rights and freedoms was brutally suppressed by the state power. The imminent fall of the communist regime in Czechoslovakia was started by a student demonstration on 17 November 1989 in Prague, which was severely confronted by the security forces. A day before this demonstration, a march of Bratislava students took place in Bratislava, on November 16th.

Political changes fundamentally affected the situation in the field of Salesian education, too. After the fall of the regime, Salesians had two options for how to complete their studies: either in the secret model they were used to, or at the Roman Catholic Faculty of Theology in Bratislava. The option of studying at the faculty was mostly used by those whose residence was near Bratislava, or who had just started their studies. Salesians who completed their studies illegally did not have formal proof of completion of their studies. In 1996 and 1997, those who needed such a document (e.g., for teaching religious education in schools) could take a state exam at one of the theological faculties.

For the sake of completeness, a list of Salesians who studied theology during the socialist period at the Roman Catholic Faculty of Theology in Bratislava[38], that is, officially, but who were secret Salesians, is also presented here: Štefan Kovalík (1967–1972), František Chudý (1968–1971), Juraj Kyseľ (1969–1973), Vojtech Surový (1969–1973), Jozef Dvorský (1973–1978), Jozef Žembera (1984–1989) (see Adamčíková 2020, pp. 43–44). The last non-public priestly ordination of Salesians took place in April 1990, when Bishop Korec ordained five Salesians (Milan Fulla, Milan Ballo, Anton Červeň, Marián Bielik and Ľudovít Baňovič).[39] Some students listed in the tables were ordained publicly in the following years (1991, 1992), after the fall of the communist regime.

## 4. Discussion of the Research and Its Limitations

According to our findings, members of the Salesian society started secret education as early as 1950 in Podolínec. However, this was more about looking for possibilities as to how to continue studies that had already begun. The first concept of complete secret philosophical and theological studies was proposed by Ernest Macák in 1951/52; it was a study of philosophy, and a year later, a follow-up study of theology. Macák was a key person in organizing the study until his arrest; he had permission for this activity from the deputy provincial Valábek. We have not been able to discover who assigned Salesians the teaching of individual subjects during the period of Provincial Bokor. In the 1950s, the harshest persecution of the churches occurred; thus, the assignment was only a matter of oral instructions, and, apparently, nobody wanted to record it, even in their notes. The model of formation and education in the 1950s and 1960s was individual. Salesians could not pursue the typical community way of life, but they lived in the so-called territorial way: they lived individually and had their superior in a wider geographical area. Naturally, it significantly complicated communication and information transfer, as well as their formation and education. It was the most difficult period; there was uncertainty, no one knew in what way the political situation would evolve and how long the persecution of the churches would last. Their concern was mainly about getting oriented in the situation, searching for a model of life in limited conditions under the threat of persecution and even imprisonment. It can be concluded, in a simplified manner, that processes were being set up, the goal of which was for the Society of Salesians in Slovakia to survive at all. A limitation in the analysis of the period of Provincial Bokor (1950s and 1960s) was the fact that we could only use memoir literature (especially by Ernest Macák, published after 1989), or secondary literature.

For the period of the other two provincials, Dermek and Kaiser (1970s and 1980s), in addition to memoir literature, we also used the research by Adamčíková (2020), based on oral history, and three interviews we conducted with living Salesians who experienced illegal study. The interviews helped us to reconstruct the structure of education during the period of normalization, when all the Salesians were gradually released from prison. However, political liberalization did not last long, and it was clear that the regime would not fall that easily and that education must be planned for the long term. In this period, clearly, it was the provincials and the Provincial Council who decided on the method of the formation and education of Salesians and the provision of individual subjects. Even though, due to security reasons, they could be involved only minimally, they were key figures who gave consent to all important activities. The life and education of Salesians in this period gradually transformed into the form of the so-called ambulatory communities. Mostly, two to four Salesians lived together in one apartment and tried to live a community life as much as possible. It was difficult to harmonize their lives, since they also studied at other universities in parallel, or they worked. It is obvious that during these two decades Salesians had already had experience with secret formation and study, and the regime was more relaxed than in the 1950s. This is also confirmed by the number of ordained Salesian priests: whereas between 1959 and 1968 there were 23 Salesians secretly ordained, between 1968 and 1990 there were 53 of them.[40] The possibilities of the secret study and its organization were related to a number of factors, especially of an external nature, several of which have been mentioned above (lack of professors and literature, problems with combining studies and work, etc.) In addition, there was the constant threat and fear of discovery, interrogation, imprisonment and torture, which, of course, created totally specific conditions.

A question may be posed: Did such study make sense? According to our findings, it appears it did. In spite of the fact that Salesians themselves do not consider their studies sufficiently extensive and of high quality, the secret study provided them with necessary knowledge to ensure their primary goal in the totalitarian society: to educate new generations of Salesians who would continue building the Salesian society.[41] This is also proven by the quantitative statistics regarding the number of Salesians in Slovakia (Table 3). It shows how their number decreased at the end of the 1960s, which can be considered the result of the regime's harsh interventions in the 1950s, and then vice versa, how it began to rise during the period of normalization, when the Salesians already had secret formation and education processes set up.

**Table 3.** Number of Slovak Salesians in selected years in the second half of the 20th century.

|  | 1953 | 1969 | 1978 | 1990 |
|---|---|---|---|---|
| **In Slovakia** | 199 | 121 | 136 | 148 |
| **In exile** | 69 | 48 | 48 | 46 |
| **In missions** | 22 | 36 | 33 | 21 |
| **Total** | 290 | 205 | 217 | 215[42] |

Despite many positives, several problems and particularities that make education of Salesians in a totalitarian regime different to regular education in free democratic conditions need to be noted. We attempt to specify them based on our existing findings. Each of the particularities has its positives and negatives, and it would be beneficial to provide their deeper analysis in the future.

If we were to evaluate only the quality of formation and education of Salesians during the period of socialism, it can be concluded that it took place under very demanding conditions (without functioning traditional Salesian educational institutions, with missing religious communities and proven ways of forming new members, etc.) and it was carried out in a minimalist mode. It was also impossible to talk about a deeper study of theology,

theological reflection, or distinguishing individual theological schools related to the application of different approaches in pastoral care. On the other hand, Salesians lived as lay people, integrated into ordinary life and work, which provided them with very good practical experiences and eliminated the risk of "detachment from reality". Their formation and education were carried out on the basis of personal contacts and trust, which shaped them morally and subsequently influenced the manner of their pastoral activity. These were new practical experiences in the field of psychology, anthropology and spirituality, which could be called practical or lived theology.

Another particularity was the motivation to become a Salesian, especially in those who entered the religious order during the socialist period. Apparently, difficult times and persecution of Churches eliminated those who would have joined the religious order for selfish, property reasons or reasons of power. On the contrary, difficult access to the Society (only through verified personal contacts) and the threat of persecution purified these external motivations. Responsible Salesians informed the prospective candidates about security risks; thus, they knew what was at stake. On the other hand, the vision of adventure, heroism, etc. could serve as motivation, too. After the fall of communism, many Salesians had trouble fitting into the communities and normal life of the Society; the aforementioned strong motivational dimension was lost.

Another particularity is represented by the simplified distinguishing situation: during the period of socialism, the external enemy was evident, all decisions and distinguishing took place in a situation and atmosphere of tension and fear of intervention of the regime. However, this often resulted in a schematic evaluation of situations and risks, without a deeper theological reflection and perception of the wider context. However, nothing else was possible at that time. The problem arose after the fall of the regime, when suddenly there was not only one external enemy, but the situation became less clear and opaque. It was necessary to re-evaluate the old ways of decision-making that were valid during the period of socialism.

From our perspective, it is necessary to more deeply analyse the personal approach and the phenomenon of trust connected to responsibility. These were the key elements of life in every order that represented a necessary condition for its survival and success. This is also confirmed by research into the Lay Apostolic Movement[43] during the socialist period, which operated on the basis of a system of small communities, based on personal relationships and trust (see Kudláčová 2019; Kudláčová and Šebová 2020). This personal approach, which was also the basis of the entire system of formation and education of young Salesians, was of great importance from a psychological point of view and was very successful. It shaped the human being in all aspects, comprehensively. Individual participants recall this feature in personal interviews in a very positive manner.

In conclusion, it can be stated that the given area has not been comprehensively explored so far, and the question is whether it is possible to reconstruct it completely, especially given the limitations of memory media, which is typical of totalitarian regimes. Nevertheless, we are convinced that further research in this area will help to reconstruct the life, formation and education of Salesians during the period of non-freedom more and more truthfully.

## 5. Conclusions

Salesians were among the most engaged religious communities in the period of the totalitarian regime in Slovakia. The same applies to their activities in organizing philosophical and theological studies. They were able to orientate themselves in the socio-political situation relatively quickly and anticipate new situations well. Each of them, whether they were priests, brothers, sisters or students, took risks and could be convicted and imprisoned for their activities. Nevertheless, Salesians were not indifferent to their vocation and were willing to undergo these dangers. They took many risks, and they could not foresee whether carrying out the secret activity would cost them their freedom, life, or whether it even made sense. Back then, they could not know what we know today.

Nevertheless, difficult periods of churches are a great paradox. They bear their fruits at the given time, but they can be identified and described better only later, from a certain distance, when they can be evaluated more objectively (for example, in the first centuries, when Christianity was not a permitted religion in the Roman Empire, the persecution of Christians paradoxically caused an increase in their number[44]). A similar finding is also confirmed by the results of our research.

Oto Mádr[45], a Czech Catholic priest and theologian, outlined the theology of the dying Church in Czechoslovakia in his essay Modus moriendi cirkvi (tr. Modus Moriendi of Church) in a rather realistic way. He saw it at the stage of extinction, dying, and he asked what God expects of those who belong to him when the Church is dying. He replies that it is essentially the same thing as at every stage: "service at its fullest" (Mádr 1992, p. 241). Mádr also tried to define particularities of the phase of dying: (1) To accept death! Absolutely reject only one manner of the Church's death: its own betrayal. (2) To live intensely! To mature to the greatest purity of service without prospects, without personal goals, to be faithful in the true form. (3) To give your best! A question must be posed; what kind of legacy will those who, perhaps after some time, will establish the Church in this space again be able to build on? (4) Not to die! This means not wanting to let the Church die, which seems to be in contradiction with the first particularity; however, the acceptance of death, even in an individual, often means mobilizing internal resources and gaining the strength for healing, or prolonging one's life (ibid.). Comparing Mádr's last will of the Church in Czechoslovakia with its fruitful period in times of persecution, from the distance of several decades, we face an incredible paradox: the Church was able not only to survive, but it raised new generations of Christians, either friars, priests, nuns, or laypeople. Based only on deeper research and analyses, which are no longer just memories, but scientific studies, it is possible to reconstruct and see everything that was going on in the Church and with the Church in the given period. The forty-year-long pilgrimage of Salesians "underground" proved to be vital and prolific, especially thanks to the courageous, active and creative activity in the field of personal accompany, formation and education of new members. This is also confirmed by Katherine Verdery (1996), who concludes that it is possible to maintain personal integrity also in a "cage" and to create small islands of freedom there.

**Author Contributions:** The authors contributed equally to this paper. Conceptualization, B.K. and A.R.; methodology, B.K.; formal analysis, B.K. and A.R.; investigation, B.K. and A.R.; resources, B.K. and A.R.; writing—original draft preparation, B.K. and A.R.; writing—review and editing, B.K. and A.R.; supervision, A.R.; All authors have read and agreed to the published version of the manuscript.

**Funding:** This research was funded by the project VEGA No. 1/0106/20 *Freedom versus Unfreedom in Education in Slovakia in 1948–1989,* supported by the Ministry of Education, Science, Research and Sport of the Slovak Republic.

**Institutional Review Board Statement:** The study was conducted in accordance with the Declaration of Helsinki, and approved by Commission for Research Ethics and Integrity, Faculty of Education, Trnava University in Trnava (No. KEIV3/2023 and 20 November 2022).

**Informed Consent Statement:** Written informed consent has been obtained from the interviewed people to publish this paper.

**Conflicts of Interest:** The authors declare no conflict of interest.

## Notes

1  The First Czechoslovak Republic was established on 28 October 1918, and its duration ended with the signing of the Munich Agreement on 30 September 1938. It was annulled after the Second World War. Under international law, it was void from the start because it was concluded under the threat of war and the use of force.

2  In Slovakia, up to 1945, Hlinka's Slovak People's Party, which brought together mainly Catholics, played an important role. Due to its collaboration with Nazi Germany during the Second World War, it was banned in 1945 and most of its leaders were convicted. Between 1946 and 1948, a part of its members continued their political activity in the Democratic Party, which primarily united Evangelicals, and some emigrated to the West.

3  In February 1948, Czechoslovakia was the last state in the Soviet sphere of influence (as essentially agreed at the Yalta Conference in 1945) where the Communists were not yet dominantly in power.

4  According to the population census of 1 March 1950, 3,432,638 inhabitants (99.72%) were registered with Churches in Slovakia, and only 9,679 (0.28%) of inhabitants stated that they were of no confession. The largest and most influential was the Catholic Church of the Roman and Greek rite (76.20% of the population claimed to be Roman Catholic and 6.55% claimed the Greek Catholic confession, for a total of 82.75%). The second most numerous Church was the Evangelical Church of the Augsburg confession (12.88% of the population) (Pešek and Barnovský 1997, p. 13).

5  Letz (2021, pp. 169–70) states that based on the data from rehabilitations from 1990, 26,079 persons were convicted in Czechoslovakia by Act No. 231/1948, of which 20,961 were in Bohemia and 5118 in Slovakia. However, there were certainly more convicted persons, because some were still tried under the old Act on the Protection of the Republic No. 50/1923 in terms of the principle that a criminal offense is assessed according to the act in force at the time when the offense was committed. By this act, a total of 5187 persons were sentenced after 1948, of which 3899 were in Bohemia and 1288 in Slovakia.

6  School Act No. 95/1948, §2.

7  *"I promise on my honour and conscience that I will be loyal to the Czechoslovak Republic and its people's democratic establishment and that I will not do anything that would be against its interests, security and integrity. As a citizen of a people's democratic state, I will conscientiously carry out my duties arising from my position, and I will do my best to support building efforts aimed at the well-being of the people."*

8  This intervention is also referred to as Barbarian Night; we use this term further on in the text.

9  See František Vnuk, *Akcie K a R: Zásahy komunistického režimu proti reholiam v rokoch 1950–1956.* Bratislava: RKCMBF (1995).

10  Technical auxiliary battalions [orig. PTP] were units of the Czechoslovak People's Army founded in the autumn of 1950. They gathered politically unreliable citizens, who were used by the state for difficult and dangerous work.

11  Monster trials in the 1950s were trials, promoted in the media, that were fabricated or did not follow the rules of a just trial. Their goal was to intimidate opponents of the regime and influence the political atmosphere in the country.

12  These processes are documented in detail in a two-volume publication by František Mikloško, Gabriela Smolíková, and Peter Smolík, eds., *Zločiny komunizmu na Slovensku 1948:1989* (tr. *Crimes of Communism in Slovakia 1948:1989*), Parts 1 and 2 (Prešov: Michal Vašek Publishing House, 2001). The first part processes individual communist persecutions chronologically. The second volume is divided into three parts: the first contains personal testimonies of persecuted people, the second contains an annotated bibliography from the field of scientific and memorial literature related to the period of communism in Slovakia, and the third part consists of lists of persecuted people. Among other interesting publications that map and document citizens persecuted for their faith are, e.g.: Veronika Lagová et al., *Smrť za mrežami* (tr. *Death behind the Bars*) (2006). Prešov: Michal Vašek Publishing House presents stories of people who were imprisoned for their faith and eventually died for it; publication, Stanislav Dzurjanin, ed. *Život za mrežami* (tr. *Life Behind the Bars*) (2007). Prešov: Michal Vašek Publishing House, presents stories of people who were imprisoned for their faith, but survived. However, even after being freed, they continued to live as if behind invisible bars, under the supervision of the ŠtB. The crimes of communism in Slovakia are mapped in detail by the Nation's Memory Institute (orig. Ústav pamäti národa) in Bratislava, which has published numerous publications on this issue; it carries out research and organizes conferences (see www.upn.gov.sk, English version https://www.upn.gov.sk/en), accessed on 25 March 2023.

13  After February 1948, 47 religious magazines were cancelled in Slovakia, among them the magazine Saleziánske zvesti (tr. Salesian News) (Kaplan and Tomášek 1994, pp. 163–64). Only the weekly Katolícke noviny (tr. Catholic Papers) and the magazine Duchovný pastier (tr. Spiritual Shepherd) remained.

14  See Jan Pešek and Michal Barnovský, *Štátna moc a cirkvi na Slovensku 1948–1953* (Bratislava: VEDA, 1997); and also Pavol Jakubčin, ed., *Likvidácia reholí a ich život v totalite* (Bratislava: Ústav pamäti národa, 2010).

15  The Order of Salesians with the official name Society of Saint Francis of Sales (the Salesian Society) was founded by St. John Bosco (real name Giovanni Melchiore Bosco, 1815–1888) in 1859 in Turin, Italy. Salesians spread very quickly to all continents and today they are the second most numerous male religious order in the world.

16  Macák's writings *Zápisky spoza mreží* (1996) (tr. *Notes from behind the Bars*) was created as a private secret diary, intended to record information about the events in the province, the liquidation of his order and the persecution of its members. The book is an authentic testimony of the life of the province in the first years of the socialist period.

17  Among them was Ernest Macák, who was sent to perform priestly work in the parish of Vysoká pri Morave. Due to the closeness of the border with Austria, Macák provided the first illegal crossings of young Salesians abroad to study theology. These were the first three smaller expeditions that took place between December 1949 and February 1950 (Macák 2006, p. 24; Adamčíková 2020, pp. 45–46).

18  Fraternity brothers have religious vows, but they do not have priestly ordination.

19  The "internment monasteries" or "centralization monasteries" were the original complexes of former cloistered religious orders, which the communist power turned into prison centers, similar to concentration camps.

20  The "Youth Building" was a symbolic name for buildings (especially between 1947 and 1957) on which young people worked. The communist regime supported and promoted these buildings, because they were supposed to demonstrate the enthusiasm

for building and dedication of young people to socialism, and at the same time, the building sites served as an instrument for influencing them.

21　(Brichta 2010), Ján. Interview, 4 April 2010.

22　For example: František Sersen, professor of moral theology, Andrej Dermek, professor of theology and pedagogy, Ernest Macák, professor of philosophy and psychology. Sersen was put in solitary confinement for this activity and later transferred to the monastery in Báč, where the provincial Bokor and his vicar Valábek were already interned.

23　Macák writes about his two-year secret service between 1950 and 1952 in his book *Dva roky v katakombách* (tr. *Two Years in the Catacombs*) (2000). He completed the manuscript in 1957.

24　Zeman's first successful escape across the Moravia River took place on 31 August–1 September 1950. During this expedition, Zeman transferred six clerics to study in Italy and at the same time requested permission for this activity from Pietro Ricaldone (the superior of Salesians). During the second expedition (22–23 October 1950), Zeman transferred 17 clerics and priests to Italy. The third, unsuccessful expedition took place on 8–9 April 1951 (see Adamčíková 2020, pp. 45–47).

25　He depicted his life in this period in the book *Živého ma nedostanete* (tr. *You Won't Get Me Alive*) (Prešov: Vydavateľstvo Michala Vaška, 2018).

26　In the 1950s, all theological faculties in Slovakia were centralized into a single one—the Roman Catholic Theological Faculty in Bratislava. The regime established the so-called numerus clausus, which limited the number of admitted students to about 32 per year, which made the chance of admission very low.

27　"Don Macák could legitimately accept temporary and lifelong religious vows from young Salesians on behalf of superiors, he could organize secret philosophical and theological studies, open a valid secret novitiate, create regional communities and appoint responsible leaders (directors) to lead them." (Bližinský 2008, p. 53).

28　Considering professional literature, there were several options: there were books published before the introduction of censorship, some teachers used their study notes as study material, some Salesians wrote their own scripts; and scripts from the Roman Catholic Faculty of Theology in Bratislava, which were available to be bought, were often used, as well as samizdat literature. In 1963, the Slovak Institute of St. Cyril and Methodius in Rome, which was the main religious and cultural institution of Slovaks abroad, was established. Over three million books were printed there, intended mainly for the Slovak secret Church; they were smuggled into Slovakia mainly during the period of normalization (Rajský 2020).

29　He depicted this experience in a book of notes, *Diagnóza: bláznom pre Krista* (Macák 2004). He had already started writing the book in 1957 under the impetus of the provincial Valábek.

30　On this occasion, Ján Beňo said to Gróf, "Ivan, this is the beginning of the growth of our congregation. Until now, I have only been dismissing brothers from the congregation who were leaving." (Peciar and Peciarová 2007, p. 10).

31　SDB Archive (n.d.), File of Jozef Izakovič, *Pamäti* (tr. *Memoir*) (typewritten), unprocessed fund, Izakovič, 18.

32　During this amnesty, 5601 people were released into civilian life, out of which 85 were friars. However, 2985 political prisoners remained in prison (Šabo 2010, p. 35).

33　Normalization means returning to the previous state (before 1968) that was considered normal by the communists. The process of normalization was connected to the period of the liquidation of the outcomes of the Prague Spring contained in the so-called Moscow Protocol from August 1968 up to the 14th Congress of the KSČ in May 1971; in a broader sense, it relates the period from August 1969 up to the fall of the regime.

34　Baranyai, Ladislav. Interview, 8 August 2020.

35　Baňovič, Ľudovít. Interview, 29 October 2020.

36　Source: Adapted from Adamčíková (2020): Štúdium teológie slovenských saleziánov v období prenasledovania cirkvi v Československu.

37　Source: Adapted from Adamčíková (2020): Štúdium teológie slovenských saleziánov v období prenasledovania cirkvi v Československu.

38　Salesians who studied theology in the period of communism abroad are not within the scope of the study.

39　Baňovič, Ľudovít. Interview, 29 October 2020.

40　Private Archives of Peter Timko (1953–1990), *Rekonštrukcie štatistík slovenských saleziánov v rokoch 1953–1990*.

41　A comparable model of education during the period of socialism in Slovakia can also be found in the Society of Jesus (Jesuits). See e.g., Ladislav Csontos, *Watersource from the Rock. Slovak Jesuits' Testimony of Faith in the Times of Communist Persecution* (Warszawa: Wydawnictwo Naukove Collegium Bobolanum, 2021).

42　Source: Private Archives of Peter Timko (1953–1990), Rekonštrukcie štatistík slovenských saleziánov v rokoch 1953–1990.

43　The lay apostolic movement was part of the secret Church during the period of socialism in Slovakia and it was led by laypeople who cooperated with secret religious, especially Jesuits and Salesians. It was aimed at working with three target groups: university youth, families and children. Its goal was to raise new generations of Christians in a totalitarian regime and preserve Christianity.

44　The number of Christians during the period of Emperor Constantine after the issuance of the edict of tolerance (313) is estimated at 5–8% (Marek et al. 2008, p. 236).

45    Mádr wrote the essay during normalization and published it for the first time in 1977 in German under a pseudonym in the magazine *Diakonia*; it was published in the Czech language in the Roman magazine *Studie* in 1980. A contemporary Czech theologian, Tomáš Halík, believes that the theology of dying matured in Mádr as early as the 1950s, when he was imprisoned and sentenced to life imprisonment.

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
