# Peer review of "How to Survive in a Totalitarian Regime: Education of Salesians in Slovakia in the Period of Socialism (1948–1989)"

_religions, doi:10.3390/rel14070858_

Round 1

Reviewer 1 Report

Summary

The paper examines the situation, survival strategies and educational opportunities of the Salesian order during the Communist totalitarian period in Slovakia.

A very detailed introduction describes the socio-political development between the state and church in former Czechoslovakia and the successor states Czech Republic and Slovakia, together with the opportunities and living conditions of the large Christian majority in Slovakia.

This is followed by the evaluation of the topic-forming question of how the Salesians survived the period of totalitarianism and persecution. The sources researched are mainly memoir literature, oral history and archives.

In properly structured chapters, the reader learns about the recruitment, study opportunities and ways of life of the novices, brothers and future priests of the order. All this in the secret, outwardly shielded underground situation in which the church, as well as the individual orders, found themselves.

Surprisingly, there were always a considerable number of students and teachers, albeit under limited conditions of content and professional competence.

The comparison of the situation within the order and the motivation of the novices to join the order during the communist persecution and illegality and the time after the political turnaround in 1989 is also revealing.

General concept

The organization and structure of the paper, the goals and the underlying research methods correspond to the four research questions. The summary and conclusions provide a consistent portrait of the researched topic.

Remarks

To start the paper with the objectives, research methods, and research state would make sense, as well as subtitling this introduction. The interested reader might also want to know why the Salesian order was chosen - are the authors members of this order?

Likewise, it would be interesting to learn how the bishops and priests of the underground church were ordained and consecrated.

The references include a good number of specialist literature from the last ten years.

Author Response

Many thanks for your rewiev and your remarks. I explain both of yours remarks in the text - the first at the p. 6 and the second at the p. 5. 

1) The Salesian order was chosen as a representative of orders that operated in Slovakia continuously in the interwar period, during the Second World War and illegally during the socialist totalitarian regime. The Salesians were numerically the largest male religious order in Slovakia in the period under review, and at the same time they belonged to the most active church communities that operated during the communist persecution.

2) The first secret bishop in Slovakia was the Jesuit Pavol Hnilica, who was consecrated by the official bishop Róbert Pobožný in January 1951. In the same year, Hnilica consecrated the Jesuit Ján Chrysostom Korec, who was the greatest moral authority in the secret church, as bishop. Most of the secret bishops were Jesuits, and as secret priests they basically ordained religious. Hnilica emigrated from Slovakia in 1952 and in 1954 presented to Pope Pius XII. and to the Superior General of the Jesuits in Rome, a report on the persecution of the Church in Czechoslovakia. The Pope responded to this news with a decree from July 1954, in which he established the exact conditions for the secret consecration of bishops and thus ensured continuous potestas ordinis, i.e. the state when at least two bishops were to be secretly consecrated in Slovakia at the same time. In general, the decree represented the basis for the establishment of secret church structures in Slovakia, and de facto remained in force until the fall of the communist regime.

Reviewer 2 Report

The text is very good, but perhaps you could state to a greater extent what was positive in the period you are dealing with.

Author Response

Many thanks for the pozitive review. It was totaliatarian regime this period, but I I mentioned some positive facts in the discussion about the research and the conclusion.

Reviewer 3 Report

I found this an innovative approach (use of memoirs, interviews, and secondary literature) to a topic that is difficult to study for the reasons cited in the paper. I am not familiar with any other literature available in English on Catholic religious orders in Eastern Europe under communism, so the paper is important.

The only real editorial suggestion I can make is to offer a little bit of general context Catholic practice and commitment in Slovakia during the communist era. Catholicism was obviously very strong in Slovakia before 1948. Did it remain so? I am not thinking of a thorough analysis but perhaps a paragraph.

Regarding some data questions, the ordinations mentioned in lines 585-586 don't fully match the lists of students in the two charts. Does this mean that some of those ordinands studied outside Slovakia? This is a small data discrepancy that a footnote might clear up. Also, the assertion in 710-712 that persecution led to increase in numbers is an empirical statement that requires a historical source to back up. 

There are a small number of places where the prose is not totally clear in English (line 400-401, 682-683). Some translations for terms are not customary in English: the word "votary," for example, usually refers to a candle, at least in American English. I believe "dress" in line 472 should be "suit" or simply "set of clothes." Forcedly, in line 294, should be forcibly. I'm not sure in English we would describe a Salesian residence as a monastery, which tends to be reserved only for cloistered religious orders. Terms that are not well known in the US context at least and require explanation or context: Barbarian Night in line 281-282, numerus clausus in line 376-377, PTP in lines 355 on. In line 337, it should be "would not" rather than "don't," and "needed" rather than "need" in line 337. I hope that helps!

Author Response

Thank you for the very precise review, grammatical improvements and also content comments. I accept all of them and have incorporated them. There would not have been many questions if 25 footnotes (from No. 15) had not fallen out during the editing of the text, where all the terms you are asking about, as well as some others, were explained. I added two more completely new notes.